

# Remote care instruction *via* the WeChat platform for female patients receiving subcutaneous anticoagulation during the COVID-19 pandemic: a retrospective analysis

Chao Yun Jiang, Ci Juan Li, Rong Zhang, Tian Hong Cai and Teng Hui Zhan

Vascular Surgery & Interventional Medicine, Fujian Maternity and Child Health Hospital, Fuzhou, Fujian, China

College of Clinical Medicine for Obstetrics & Gynecology and Pediatrics, Fujian Medical University, Fuzhou, Fujian, China

## ABSTRACT

**Background**. The purpose of this study was to estimate the effect of remote nursing guidance based on WeChat platform for female patients receiving subcutaneous anticoagulation during the COVID-19 epidemic.

**Methods**. Retrospective analysis of clinical data, including demographic data and anticoagulation complications, of 126 female patients who received subcutaneous anticoagulation therapy and received remote nursing guidance using WeChat platform in our hospital from January 2022 to December 2022. The Anti-Clot Treatment Scale (ACTS) and the World Health Organization Quality of Life-BREF (WHOQOL-BREF) scale were used to evaluate patients' satisfaction with anticoagulation and quality of life at the beginning of anticoagulation, half a month after anticoagulation, and after three months of anticoagulation.

**Results**. In total, 126 patients were involved in this study, all of them were female, 115 cases were natural pregnancy, 11 cases were assisted reproduction. This study included seven cases of lower extremity deep vein thrombosis, 100 cases of hypercoagulable state, 10 cases of antiphospholipid syndrome, and eight cases of protein S deficiency, one case of hyperhomocysteinemia. During the follow-up period, four patients (3.17%) had subcutaneous injection complications, including three cases of subcutaneous hemorrhage and one case of liquid leakage. A total of 123 patients had completed the planned anticoagulation therapy or were receiving anticoagulation therapy as planned, and three patients did not receive anticoagulation therapy as planned (zero cases lost contact, two cases changed treatment units, and one case refused treatment). ACTS score (55.03 ± 1.73) and WHOQOL-BREF score (62.18 ± 3.17) after three months of anticoagulation, ACTS score (54.18 ± 1.20) and WHOQOL-BREF score (60.37 ± 2.25) after half a month of anticoagulation was significantly higher than the ACTS score (47.81 ± 1.69) and WHOQOL-BREF score (55.25 ± 1.85) at the beginning of anticoagulation, and the difference was statistically significant ($P$ value < 0.01).

**Conclusions**. During the COVID-19 pandemic, remote nursing instruction *via* the WeChat platform for female patients receiving subcutaneous anticoagulation can increase anticoagulation compliance, satisfaction, and quality of life.

Corresponding author
Teng Hui Zhan,
zth182999@fjmu.edu.cn

## INTRODUCTION

The novel coronavirus disease (COVID-19) first appeared in Wuhan, China, and quickly spread around the world in December 2019 (*Gates, 2020*). On January 30, 2020, it was announced as a global public health emergency of international concern (*Garfin, Silver & Holman, 2020*). Despite many measures taken to control the virus, a large number of people are still infected. Some of them have died due to ineffective treatment, while others have died from complications arising during or after the infection (*Huang et al., 2020*). Quarantine measures to reduce social activities and going out are considered to be effective in preventing and controlling the epidemic (*Xu, Wu & Cao, 2020*). In this situation, it is difficult to go out for medical treatment, and there is also a risk of COVID-19 infection when going out, which will undoubtedly have a huge impact on some patients with diseases that require close follow-up.

Lower extremity deep vein thrombosis, hypercoagulable state, antiphospholipid syndrome, protein C deficiency, protein S deficiency, hyperhomocysteinemia and other diseases require anticoagulant therapy (*De Maeseneer et al., 2022*; *American College of Obstetricians & Gynecologists' Committee on Practice Bulletins-Obstetrics, 2018a*; *Royal College of Obstetricians Gynaecologists, 2015*; *American College of Obstetricians Gynecologists Committee on Practice Bulletins-Obstetrics, 2018b*). Commonly used anticoagulant drugs include warfarin, unfractionated heparin, low molecular weight heparin, and new oral anticoagulant drugs. Anticoagulant methods include oral, intravenous, and subcutaneous injections (*De Maeseneer et al., 2022*). Most of the female patients diagnosed with the above diseases were in the perinatal period. In order to avoid the impact of anticoagulant drugs on the fetus, they had to be anticoagulated by subcutaneous injection of low molecular weight heparin (*Queensland Health, 2023*). Incorrect subcutaneous injections may lead to risks such as bleeding, making it necessary for them to receive regular outpatient care guidance (*Yan, Wei & Ping, 2019*). With the outbreak of COVID-19, these patients cannot get help from nursing staff in time. They are not only concerned about the risk of COVID-19 infection on the way to the hospital, but also about the occurrence of complications and treatment delays. Therefore, it is crucial to provide remote nursing guidance for patients undergoing subcutaneous anticoagulation during the COVID-19 epidemic.

The WeChat platform is the most widely used social software in China, with at least one billion people using WeChat every day (*Shao et al., 2020*). According to reports, WeChat can serve as a remote medical model for monitoring, preventing, and managing diseases such as infertility, chronic diseases, and infectious disease (*Zhang, Xiao & Chen, 2019*; *Jiang et al., 2020*; *Feng et al., 2017*). In 2023, a study conducted during the COVID-19 pandemic demonstrated that virtual visits (VVs) for cardiac electrophysiology patients were as feasible and effective as in-person visits, with high patient satisfaction (*Mariani et al., 2023*). While the WeChat platform offers various communication methods, such

as text, voice, and video, these cannot fully replace face-to-face interactions. Nonverbal cues like facial expressions, tone of voice, and body language may be missed, impacting the understanding and trust between doctors and patients. Additionally, subcutaneous injections are medical procedures, and remote guidance lacks the direct observation and assessment possible in person, potentially affecting the accurate evaluation of treatment effects. Furthermore, remote nursing guidance involves transmitting and storing patients' health information and images of private areas over the network, raising significant security concerns. Any data breach could lead to privacy violations and legal issues. Thus, there are considerable challenges in determining whether remote nursing guidance *via* the WeChat platform can truly benefit patients undergoing anticoagulation therapy.

During the COVID-19 epidemic, we utilized WeChat platform to offer remote care guidance and follow-up for patients receiving subcutaneous anticoagulation. This study summarizes our experience and clinical outcomes, providing a basis for remote nursing guidance and insights for future telemedicine research.

## METHOD

Portions of this text were previously published as part of a preprint (https://doi.org/10.21203/rs.3.rs-3268863/v1).

### Ethics approval and consent to participate

This study was approved by the ethics committee of Fujian Maternity and Child Health Hospital and strictly adhered to the tenets of the Declaration of Helsinki (Code of Ethical approval for scientific research project: no. 2023KY013). In addition, all patients and their guardians signed an informed consent form before deciding to participate in the study.

### Patient

The data of 126 patients who received subcutaneous anticoagulation in Fujian Maternity and Child Health Hospital, Affiliated Hospital of Fujian Medical University from January 2022 to December 2022 and used WeChat platform for remote nursing guidance were retrospectively analyzed (deriving from electronic health records, surveys, and medical records from the hospital).

Inclusion criteria included the following: (1) Adult female patients aged 18 years and older; (2) Diagnosed with diseases requiring anticoagulation therapy; (3) Currently receiving or planning to start subcutaneous anticoagulation therapy; (4) Possessing a smartphone and proficient in using WeChat; and (5) Willing to participate in the study and provide informed consent.

Exclusion criteria included the following: (1) Combined with other serious diseases, requiring additional treatment, (2) Incomplete data, (3) Inconvenient to use the Internet, (4) Refused to participate in this study.

### Remote nursing guidance based on WeChat platform
#### *Training of team members*

The team consisted of one vascular surgeon and three vascular surgery nurses, all of whom were health care professionals (doctors and nurses). They managed the WeChat platform

and provided remote nursing guidance to patients. Prior to conducting remote nursing guidance, all team members received comprehensive training to ensure mastery of necessary skills. This training included the use of the WeChat platform, specialized knowledge of anticoagulation therapy, information security and privacy protection, technical support and troubleshooting, patient education and psychological support, and teamwork and communication.

### Education module and a question-and-answer (Q&A) module

The remote nursing guidance provided through the WeChat platform mainly included two parts: an education module and a question-and-answer (Q&A) module. The educational modules focused on daily health education for patients. Graphic materials about anticoagulation therapy were posted in the group announcement section on the WeChat platform. These materials included popular science knowledge about various related diseases, types of anticoagulant drugs, precautions during anticoagulation, procedure of subcutaneous injection and treatment of complications. Patients could view and learn these materials at any convenient time through the group announcements, especially if they had questions about anticoagulation therapy or subcutaneous injections. The content of the group announcement was regularly updated, occasionally including treatment success stories. In the Q&A module, one medical staff member was online from 8:00 to 20:00 h every day to reply to patients' questions, send reminders, urge outpatient to regularly review, and remind patient that delayed treatment may lead to consequences. Patients could choose the way of information exchange according to their preferences. Medical staff also guided patients to communicate, discuss, and share nursing experience on the WeChat platform, corrected erroneous understandings and practices, and encouraged good habits.

We guided patients to read health education content on WeChat platforms during hospitalization or outpatient visits, sent inspection data through WeChat, and asked questions on the WeChat platform to ensure that they master the correct usage methods.

## Research tools

A questionnaire was used to find out the patients' anticoagulation satisfaction and quality of life.

(1) The Anti-Clot Treatment Scale (ACTS) was improved from the Duke Anticoagulation Satisfaction Scale (DASS), including 17 items in 2 dimensions of perceived trouble, burden and perceived benefit. The scale was scored using the Likert 5-point scoring method, not at all = 5 points, a little = 4 points, moderate = 3 points, quite large = 2 points, extremely large = 1 point, and the positive questions are assigned opposite points. A higher value indicates that the patient was more satisfied with anticoagulation therapy. *Cano et al. (2012)* and other scholars conducted a cross-sectional survey on 1,336 patients who had been using anticoagulants for a long time in many countries. The verified languages were Dutch, French, German, English and Italian. The research showed that, except for the Cronbach's alpha = 0.79 and test-retest reliability of 0.72 in Dutch, the Cronbach's alpha and test-retest reliability coefficients of the perceived burden and perceived benefit items in other languages were greater than 0.82. *Yile, Xiaoling & Guihua (2021)* from China

translated ACTS into Chinese for reliability and validity verification, with Cronbach's alpha = 0.87 and a retest reliability of 0.93. Currently, ACTS was the most widely used scale for evaluating anticoagulation satisfaction worldwide.

(2) The World Health Organization Quality of Life-BREF (WHOQOL-BREF) scale was developed on the basis of the WHOQOL-100 scale (*Sijtsma et al., 2008*). The WHOQOL-BREF scale retains the comprehensiveness of the original scale. The scores of each field have a high correlation with the scores of the corresponding fields in the original scale (the correlation coefficient is between 0.89 and 0.95) (*Kalfoss & Halvorsrud, 2009*). The scale consists of 26 items, including the physiological field, psychological field, social relations field, and environmental field. Item 1 and Item 2 are two independent topics, and their total scores are used as a comprehensive index to evaluate the quality of life. Each item in the scale is designed to be graded 1-5, corresponding to 1-5 points, and items 3, 4 and 26 are scored in reverse and graded 1-5, corresponding to 5-1 points. The higher the score, the higher the quality of life.

### Research process

In our daily outpatient practice, we use the WHOQOL-BREF and ACTS scores to assess quality of life and adherence in anticoagulated patients, which are recorded in their medical records. We believe that visualizing changes in these scores can boost patients' confidence and encourage adherence to their treatment, which can be a prolonged process.

We designed a retrospective observational study protocol and obtained ethical approval. Participants were included through outpatient clinics and WeChat. An announcement was posted in the WeChat platform in March 2023 to inform participants of the study. Participants who had used the WeChat platform from January to December 2022 were eligible to participate. A total of 126 participants responded and provided informed consent to participate. After obtaining informed consent, we collected data, including ACTS and WHOQOL-BREF scores. The collected data were then analyzed, and the results were interpreted to form the study's findings.

Study participants did not experience an increased healthcare burden or a change in their treatment regimen due to their participation in this study.

### Data acquisition

The researchers collected relevant data such as the demographic characteristics of the patients. ACTS scores and WHOQOL-BREF scores were collected at the beginning of anticoagulation, half a month after anticoagulation, and 3 months after anticoagulation; whether anticoagulation is on schedule, and whether there are complications such as subcutaneous bleeding, leakage, allergies, bent needles, and broken needles.

### Statistical analysis

SPSS 21.0 software was used for statistical analysis. Continuous data were presented as mean ± standard deviation and range. A routine distribution test histogram and Shapiro–Wilk test in nonparametric tests was performed for all continuous data, and the data were confirmed to have a normal distribution. Clinical parameters between the two groups

**Table 1  Demographical and clinical characteristics of the patients in this study.**

| Item | Number |
|---|---|
| Number of patients | 126 |
| Age (year), mean ± SD | 31 ± 4.5 |
| Gender of patients (%) | |
|     Male | 0 |
|     Female | 100% |
| Natural pregnancy | 115 |
| Assisted reproduction technologies (ART) | 11 |
| Disease | |
|     Deep vein thrombosis (DVT) | 7 |
|     Hypercoagulable state | 100 |
|     Antiphospholipid syndrome (APS) | 10 |
|     Protein S deficiency | 8 |
|     Hyperhomocysteinemia | 1 |

Notes.

This table summarizes the demographic and clinical characteristics of the female patients included in the study. The total number of patients is 126, with an average age of 31 years. All patients are female. The majority of the pregnancies were natural (115 cases), with 11 cases involving assisted reproduction technologies (ART). The clinical conditions observed among the patients include deep vein thrombosis (DVT) in seven patients, a hypercoagulable state in 100 patients, antiphospholipid syndrome (APS) in 10 patients, protein S deficiency in eight patients, and hyperhomocysteinemia in one patient.

were compared with the independent samples $t$-test. A $p$-value of <0.05 was defined as significant.

## RESULT

Our medical institution treats approximately 2,000 anticoagulation patients annually, of which about 1,800 do not meet the inclusion criteria for the study. The main reasons are as follows: (1) Did not use the WeChat platform for remote nursing guidance. (2) Have other comorbidities and are undergoing treatments such as pregnancy preservation. (3) Unwilling to participate in clinical research.

This study included 126 patients, all of whom were women, with a median age of 31 years old and a range of 20–46 years old, 115 cases were natural pregnancy, 11 cases were assisted reproduction. Among them, there were seven cases of deep vein thrombosis of lower extremities, 100 cases of hypercoagulable state, 10 cases of antiphospholipid syndrome, eight cases of protein S deficiency, and one case of hyperhomocysteinemia (Table 1).

During the follow-up period, a total of four patients (3.17%) had subcutaneous injection complications, of which three patients had subcutaneous bleeding, one patient had fluid leakage, zero patients had allergies, and zero patients had bent or broken needles (Table 2).

A total of 123 patients had completed the planned anticoagulant therapy or were receiving anticoagulant therapy as planned (97.6%), and three patients did not receive anticoagulant therapy as planned (zero cases lost contact, two cases changed treatment units, and one case refused treatment) (Table 3).

The evaluation results of the patient's ACTS and WHOQOL-BREF scale showed that: when anticoagulation was started, the patient's average ACTS score was 47.81 ± 1.69, and

**Table 2  Complications during follow-up.**

| Item | Number |
|---|---|
| Subcutaneous hemorrhage | 3 |
| Liquid leakage | 1 |
| Allergy | 0 |
| Bent needle or broken needle | 0 |

Notes.

This table presents the complications observed during the follow-up period of the study. There were three instances of subcutaneous hemorrhage, one case of liquid leakage, and no cases of allergy or bent/broken needles.

**Table 3  Follow-up outcomes.**

| Item | Number |
|---|---|
| Completed the planned anticoagulation or are receiving anticoagulation treatment as planned | 123 |
| Anticoagulation was not carried out as planned | |
|     Lost contact | 0 |
|     Change treatment unit | 2 |
|     Refuse treatment | 1 |

Notes.

This table presents the follow-up outcomes of patients undergoing anticoagulation therapy. Out of 126 patients, 123 completed the planned anticoagulation or are currently receiving anticoagulation as scheduled. Three patients did not adhere to the planned anticoagulation: two changed treatment units, and one refused treatment.

**Table 4  ACTS and WHOQOL-BREF scores at different time points during anticoagulation.**

| Item | At the beginning of anticoagulation $\bar{x} \pm s$ | Half a month after anticoagulation $\bar{x} \pm s$ | 3 months after anticoagulation $\bar{x} \pm s$ |
|---|---|---|---|
| ACTS score | $47.81 \pm 1.69$ | $54.18 \pm 1.20$ | $55.03 \pm 1.73$ |
| WHOQOL-BREF score | $55.25 \pm 1.85$ | $60.37 \pm 2.25$ | $62.18 \pm 3.17$ |

Notes.

This table summarizes the ACTS (Anti-Clot Treatment Scale) and WHOQOL-BREF (World Health Organization Quality of Life - BREF) scores measured at three time points: at the beginning of anticoagulation, half a month after starting anticoagulation, and three months after starting anticoagulation. The ACTS scores improved progressively from $47.81 \pm 1.69$ at the start to $55.03 \pm 1.73$ at 3 months. Similarly, WHOQOL-BREF scores increased from $55.25 \pm 1.85$ at the beginning to $62.18 \pm 3.17$ at 3 months, indicating an improvement in patient quality of life over time.

the average WHOQOL-BREF score was $55.25 \pm 1.85$. After half a month of anticoagulation, the patient's average ACTS score was $54.18 \pm 1.20$, the average WHOQOL-BREF score was $60.37 \pm 2.25$, significantly higher than before, and the difference was statistically significant ($P < 0.01$). After 3 months of anticoagulation, the patient's average ACTS score was $55.03 \pm 1.73$, the average WHOQOL-BREF score was $62.18 \pm 3.17$, which was higher than that at the beginning of anticoagulation and two weeks after anticoagulation, and the difference was statistically significant ($P < 0.01$) (Tables 4–5).

# DISCUSSION

In 2021, with the opening up of China's "three child" policy, more and more people choose to give birth again, resulting in an increasing number of elderly pregnant women. According

**Table 5  Comparison of ACTS and WHOQOL-BREF scores at different time points during anticoagulation.**

| Comparison items | ACTS | | BREF | |
|---|---|---|---|---|
| | *t*-value | *P*-value | *t*-value | *P*-value |
| At the beginning of anticoagulation & Half a month after anticoagulation | 45.03 | <0.01 | 49.01 | <0.01 |
| At the beginning of anticoagulation & 3 months after anticoagulation | 45.80 | <0.01 | 25.19 | <0.01 |
| Half a month after anticoagulation & 3 months after anticoagulation | 7.39 | <0.01 | 6.47 | <0.01 |

**Notes.**
ACTS, Anti-Clot Treatment Scale; WHOQOL-BREF, World Health Organization Quality of Life - BREF.
This table presents the paired *t*-test analysis comparing ACTS and WHOQOL-BREF scores at various time points during anticoagulation therapy. Significant improvements were observed in both ACTS and WHOQOL-BREF scores between the beginning of anticoagulation and half a month after, as well as between the beginning and three months after anticoagulation. The comparison between half a month and three months after anticoagulation also showed significant improvement, indicating a continuous enhancement in patient treatment satisfaction and quality of life over time.

to the population monitoring statistics of Shanghai in China, the average childbearing age in 2022 is 31.18 years old, which is more consistent with the median age of 31 years old in this study. Older age is an independent risk factor for venous thromboembolism (VTE) (*Grant et al., 2016*). In addition, with the increase of people's desire to have children, more and more people who were originally unable to have children choose assisted reproduction. According to the "China Assisted Reproduction Research Report 2023", the total number of assisted reproductive treatment cycles exceeds 1 million each year in China, and the number of babies born exceeds 300,000, accounting for approximately 3% of the total birth population. In this study, there were 11 patients with assisted reproductive technology, accounting for 8.7%. Among the hypercoagulable patients, a large number of patients were found in the screening before receiving assisted reproductive technology, and assisted reproductive technology is also an independent risk factor for VTE (*Queensland Health, 2020*). As a result, there is an increasing number of perinatal women requiring anticoagulation therapy and a growing need for remote care instruction.

Subcutaneous low-molecular-weight heparin (LMWH) is widely regarded as the optimal anticoagulant for women during the perinatal period (*American College of Obstetricians Gynecologists Committee on Practice Bulletins-Obstetrics, 2018b*). The pain and bleeding risks caused by subcutaneous injection can reduce the anticoagulant compliance of patients, and literature shows that the compliance rate is only 60% (*Guntupalli et al., 2020*). Therefore, enhancing compliance among anticoagulated patients remains a significant challenge for healthcare professionals.

In recent years, with the widespread use of smartphones, especially in China (*Mihalko, 2015*), telemedicine based on social media platforms is used for health management and disease-related education. Social media can provide healthcare advice and allows remote patient and clinician contact, care, advice, reminders, education, monitoring, and remote admission (*Liu et al., 2020*). A European study used mobile apps to promote health education, maintain antiretroviral therapy, and reduce anxiety levels in HIV patients in older adults (*Longinette et al., 2017*). A New Zealand study used a mobile app

for colonoscopy preoperative preparation (*Liu et al., 2014*). WeChat is an application widely used in China, similar to Facebook, Twitter and WhatsApp. It is a popular, convenient, interactive, and intuitive way of information exchange. Therefore, it is an important platform to maximize information coverage and effectiveness (*Lyu et al., 2016*). The WeChat platform provides a convenient and cost-effective approach for remote care guidance. In China, WeChat accounts are free, and network costs are minimal. Consequently, we utilized the WeChat platform to offer remote care guidance to anticoagulated patients during the COVID-19 pandemic.

In our study, the patients' ACTS score and WHOQOL-BREF score gradually improved, which means that anticoagulation compliance and quality of life have been improved. This study achieved a 97.6% completion rate for anticoagulation therapy, significantly higher than the 60% reported in previous studies, suggesting that remote care guidance *via* the WeChat platform is highly effective in enhancing patient adherence. High adherence is crucial for anticoagulation therapy, as it directly correlates with treatment efficacy and reduces the risk of thrombotic recurrence. Only 3.17% of patients in this study experienced subcutaneous injection complications, and there were no serious adverse events.We believe that this is due to the timely communication and nursing guidance through the WeChat platform, so that patients can be dealt with in a timely manner when encountering problems.

In 2022, a randomized controlled trial utilized the WeChat platform for telemedicine education and care guidance for parents of children with type 1 diabetes mellitus. Six months after discharge, the intervention group showed significantly higher WHOQOL-BREF scores compared to the control group (*Huang, Wang & Wu, 2022*). Similarly, a study on primiparous women undergoing cesarean delivery found that continuity of care through the WeChat platform resulted in significantly higher quality of life scores (36-item Short Form Health Survey, SF-36) and a lower rate of complications in the observation group compared to the control group (*Wang et al., 2024*). Another study involving adults with type 2 diabetes demonstrated that WeChat-based online intervention significantly improved medication adherence in the intervention group (*Zhang et al., 2024*). These findings align with those of the present study, underscoring the substantial potential of WeChat-based telemedicine services.

A gratifying phenomenon for us is that although the COVID-19 epidemic is over now, among the patients included in the study, a portion of patients who have already completed anticoagulation therapy will maintain long-term contact with us. They will consult health issues that are not related to the content of this study on the WeChat platform and also help their family or friends consult on health issues and seek help. This phenomenon has prompted us to carry out this study. At the same time, we believe that the health guidance and management methods provided by the WeChat platform are not only applicable during the COVID-19 epidemic, and we are also conducting research in this area.

This study has several limitations. First, as a single-center study with a small sample size, the results may not be generalizable to other populations, limiting the extrapolation of findings. The feasibility and effectiveness of telecare coaching may also vary across different healthcare settings. Second, the retrospective, observational nature of the study, coupled

with the absence of a control group, weakens its ability to infer causality, as potential confounders could not be fully controlled. Third, the reliance on self-reported data, such as ACTS and WHOQOL-BREF scores, may introduce biases, including recall and social desirability bias, due to the subjective nature of patient responses. Furthermore, the lack of objective measures restricts the depth of outcome assessment. Fourth, due to technical limitations, we did not collect information on engagement with the WeChat platform, such as the number of times participants viewed educational content, their visit frequency and duration, and the frequency and number of questions asked by patients during the WeChat Q&A sessions. This information could help us gain deeper insights into the role of remote nursing guidance from a new perspective. Finally, ongoing concerns remain regarding data security and patient privacy in Internet-based telemedicine services.

To address these limitations, we are conducting a prospective cohort study to minimize potential confounders and enhance the credibility and generalizability of our findings. Additionally, we will expand the study to include other high-risk patient groups, such as those with hemangiomas and vascular malformations. Furthermore, we will assess the impact of telecare guidance on long-term outcomes.

## CONCLUSION

During the COVID-19 pandemic, remote care guidance through the WeChat platform for patients receiving subcutaneous anticoagulation can increase anticoagulation compliance, satisfaction, and quality of life for patients.

### Funding
This work was supported by the grants from the Scientific Research Fund for National Capacity Building Project on Prevention and Treatment of Pulmonary Embolism and Deep Venous Thrombosis (Grant No Y101). The funders had no role in study design, data collection and analysis, decision to publish, or preparation of the manuscript.

### Grant Disclosures
The following grant information was disclosed by the authors:
The Scientific Research Fund for National Capacity Building Project on Prevention and Treatment of Pulmonary Embolism and Deep Venous Thrombosis: Y101.

### Competing Interests
The authors declare there are no competing interests.

### Author Contributions
- Chao Yun Jiang conceived and designed the experiments, performed the experiments, analyzed the data, prepared figures and/or tables, authored or reviewed drafts of the article, and approved the final draft.

- Ci Juan Li performed the experiments, authored or reviewed drafts of the article, and approved the final draft.
- Rong Zhang analyzed the data, prepared figures and/or tables, and approved the final draft.
- Tian Hong Cai analyzed the data, prepared figures and/or tables, and approved the final draft.
- Teng Hui Zhan conceived and designed the experiments, performed the experiments, authored or reviewed drafts of the article, and approved the final draft.

## Human Ethics

The following information was supplied relating to ethical approvals (i.e., approving body and any reference numbers):

The ethics committee of Fujian Provincial Maternal and Child Health Hospital approval to carry out the study within its facilities (Ethical Application Ref: 2023KY013).

## Data Availability

The raw measurements are available in the Supplementary File.

## Supplemental Information

Supplemental information for this article can be found online at http://dx.doi.org/10.7717/peerj.18337#supplemental-information.

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
