# Peer review of "Remote care instruction *via* the WeChat platform for female patients receiving subcutaneous anticoagulation during the COVID-19 pandemic: a retrospective analysis"

_PeerJ, doi:10.7717/peerj.18337_

## Round 0.1 · original submission · Major Revisions

This study is interesting but requires extensive changes. The reviewers have suggested some major revisions to your manuscript. In addition to addressing points raised by the reviewers, please also respond to the following:
1. You have stated that all study participants signed informed consent prior to the study. However, recruitment occurred between January and December 2022 and ethics approval was not granted until July 2023. Please explain this.
2. Please state where this study was registered.
3. Please state how participants were recruited.
4. Please state how many participants declined to participate in the study, or did not meet your inclusion criteria.
5. How was questionnaire data collected and by whom?
6. How were adverse events defined and how and by whom was information about them collected? Was there any structured follow up to assess for this in all study participants, or was this only if participants reported this?
7. How was the WeChat education and question and answer module developed?
8. It is stated that all team members were qualified medical staff - please confirm if this means they were all health care professionals (doctors and nurses) or if they were all doctors (medical health care professionals).
8. Please present the follow up time period in months.
9. Limitations should be in a separate section and not presented as part of your conclusion.

·

Basic reporting

Although this topic is important and addresses a strategy that can be useful in various remote care programs, the document requires important changes to be considered in a publication.

The references are sufficient and recent.

The title refers to patients who received subcutaneous anticoagulation, however, only women were studied, which is why it is necessary to modify the title in such a way that it refers to the fact that it is only a study for this group of people.

It is recommended that keywords be integrated into The Medical Subject Headings

In the objective of the study, it is recommended to change the word "investigate" to estimate.

In line 62, add the space between the last word and the reference 1.

Experimental design

The process of collecting information is not clear, it is pointed out that it was a retrospective study where information was obtained from the files and later the support by health personnel through wechat was mentioned and several instruments such as quality of life were also applied. Therefore, it is suggested to review the wording of the methodology and present a scheme that expresses the actions carried out chronologically.

The statistical analysis described is relevant to present the characteristics of the participants and the possible associations with the event of interest. It is necessary to mention what type of analysis was carried out to know the distribution of numerical variables to later select the parametric or non-parametric tests as appropriate, especially because the authors present the median age, but use student's t-test.

In addition, a multivariate analysis is required for the control of possible confusing variables and the integration of a model that allows to know the associations of the main variables studied.

Validity of the findings

The results presented do not contribute to recognizing the impact of the strategy as expected.

The discussion should focus on the variables studied and the effect that the intervention had by avoiding repeating what was found in its results, it is important to contrast its results with respect to other studies and provide an explanation of these.

·

Basic reporting

1. Basic Reporting
Strengths:
• The manuscript is written in clear and professional English.
• The introduction and background provide adequate context.
• Relevant literature is cited appropriately.
Weaknesses:
• Some sections could benefit from improved clarity. For instance, the discussion section tends to be repetitive and could be streamlined.
• Figures and tables, while relevant, could use more detailed captions to enhance understanding.
• The raw data should be included in the supplementary materials for transparency and verification purposes.

Experimental design

2. Experimental Design
Strengths:
• The study addresses an important issue within the scope of the journal.
• The research question is well-defined and relevant, aiming to fill a gap in the existing knowledge.
Weaknesses:
• There is a lack of a control group, which is crucial for comparing the effectiveness of the intervention.
• The methods section lacks sufficient detail for full replication. For example, more information is needed on the specific content of the educational modules and the training of the staff providing the guidance. Authors should provide that

Validity of the findings

. Validity of the Findings
Strengths:
• The findings are relevant and provide valuable insights into remote care during the pandemic.
• The statistical analyses used are appropriate for the data.
Weaknesses:
• The sample size is relatively small and restricted to a single center, which may limit the generalizability of the results.
• The follow-up period is short, making it difficult to assess long-term outcomes of the intervention.
• The improvement in ACTS and WHOQOL-BREF scores, while statistically significant, may not be clinically significant. The manuscript should discuss the clinical relevance of these changes in more depth.

Additional comments

General Comments
• The manuscript provides a timely contribution to the literature on remote healthcare, particularly during the COVID-19 pandemic.
• The authors should consider conducting a randomized controlled trial to strengthen the evidence base for their findings.
• Future studies should aim for a larger sample size and include multiple centers to enhance the generalizability of the results.
• The discussion should address the limitations more thoroughly, particularly the potential biases inherent in a retrospective study design and the lack of a control group.
• The manuscript is of publishable quality but would benefit from revisions to address the methodological limitations and enhance the clarity of the reporting.
• The authors should be encouraged to provide the raw data and consider a more robust study design in future research.

Suggested Revisions
1. Introduction:
o Expand the introduction to provide more detailed background on the challenges of managing anticoagulation therapy remotely.
o Moreover in order to empower the feasibility and efficacy aspect of telemedicine I would strongly suggest the authors to include and discuss: The Feasibility, Effectiveness and Acceptance of Virtual Visits as Compared to In-Person Visits among Clinical Electrophysiology Patients during the COVID-19 Pandemic. J Clin Med. 2023 Jan 12;12(2):620. doi: 10.3390/jcm12020620. PMID: 36675547; PMCID: PMC9865180.
2. Methods:
o Provide a more detailed description of the educational modules and the training of staff.
o Clarify the criteria for selecting patients and the exact nature of the interventions.
3. Results:
o Provide more detailed tables and figures with comprehensive captions.
4. Discussion:
o Streamline the discussion to avoid repetition.
o Address the clinical significance of the findings.
o Discuss limitations in greater detail and suggest directions for future research

---

## Round 0.2 · Minor Revisions

Thank you for presenting a revised version of your manuscript. There are a few outstanding minor issues that require addressing.

• In the introduction, it states: ‘Despite many measures taken to control the virus, a large number of people are still infected and some of them died due to ineffective treatment’. Many people who die with or post COVID is not due to ineffective treatment, but rather complications arising during or post infection. This cannot be solely attributed to ineffective treatment.
• The issue of informed consent has not been adequately addressed. I do not understand how a patient can sign informed consent before the study was undertaken (stated on page 5), when this was a retrospective review. Can you please clarify if patients were approached to allow permission to retrospectively access their data? In your rebuttal letter, you state that patients were not recruited but selected from existing records. However, the manuscript states that patients signed informed consent before the study was undertaken. This is contradictory and unclear in the manuscript. Please clarify this.
• It is unclear how patients were approached for study participation. As stated above, please clarify if participants were approached to sign informed consent to allow retrospective review of their data. If this was the case, it seems very unusual to not keep a record of how many declined and for what reason. Again, this is unclear from the manuscript as your rebuttal letter states that patients believed this may increase their medical care burden. However, if the study was performed retrospectively, the data had already been collected so I don’t really see how this is possible.
• It is very relevant that 1800 of 2000 patients that are seen annually for anticoagulation were not eligible for inclusion in the study. Please state why they were not eligible and report this in the manuscript.
• Do you have any information on engagement with the WeChat platform e.g. how many times participants reviewed the education content, how often this was accessed and for how long, and how often and by how many patients questions were asked during the Q&A sessions on WeChat.

·

Basic reporting

The new version of the document is well structured, delimits its content and the tables and figures comply with the established criteria.

Experimental design

The responses to the observations made by the reviewers are explained in a clear and specific way, make the suggested changes and integrate new information that allows clarifying the activities carried out in the study.

Validity of the findings

In the context of the COVID-19 pandemic, the work is relevant in exploring new care mechanisms in women receiving an anticoagulant. Its usefulness is important for patient care in adverse scenarios such as pandemics.

Doubts about the ethical aspects and limitations of the study are answered in a timely manner.

Additional comments

none.

---

## Round 0.3 · Minor Revisions

Thank you for providing a revised manuscript and clarification of your recruitment and consent process. Please include in the manuscript how participants were recruited (e.g. an announcement was posted in the WeChat platform in March 2023 to inform participants of the study. Participants who had used the WeChat platform from January to December 2022 were eligible to participate. A total of 126 participants responded and provided informed consent to participate). This will make it much clearer for readers to understand how this process was undertaken.

Please also add in to the limitations section that no information on engagement with the WeChat platform was collected.

---

## Round 0.4 · accepted · Accept

Thank you for addressing all of the editor and reviewer comments in this revised version of your manuscript.